# REINFORCEMENT LEARNING WITH PARTIAL ORDER REPRESENTATION FOR MONOTONIC PHYSICAL SYSTEM

## ABSTRACT

Prior model-free reinforcement learning techniques may struggle with complex high-dimensional visual-motor control tasks such as rope manipulation and pouring water due to the high sample complexity involved in state representation and dynamics learning. These tasks often involve physical systems that preserve the property of monotonicity, such as water always flowing downwards when poured. Motivated by this insight, we propose the Partial Order Representation (POR) framework, which improves reinforcement learning algorithms' ability to capture the systems' monotonicity, resulting in valuable signals during training that can enhance performance and reduce sample complexity. Our experiments demonstrate that the POR framework outperforms state-of-the-art methods in terms of sample efficiency and performance across a diverse set of challenging visual-motor control tasks.

## 1    INTRODUCTION

Reinforcement learning (RL) is a powerful approach for training agents to perform complex tasks through trial and error, and its success has been demonstrated across a wide range of challenging tasks. However, when it comes to high-dimensional visual-motor control tasks like pouring water from a teapot into a cup accurately(Li et al., 2022; Huang et al., 2021), realigning a rope to its original straightened form(Nair et al., 2017; Sundaresan et al., 2020), and other similar endeavors(Zhu et al., 2021; Salhotra et al., 2022; Lin et al., 2022a), model-free RL may encounter difficulties. These tasks demand advanced state representation and dynamics learning with high sample complexity.

In fact, many systems possess some implicit physical properties that can be highly beneficial for learning algorithms but have been overlooked by previous work, such as the **monotonicity** of the environment. **Monotonicity** (Figure 1), the property of a function or system state variable that consistently and steadily increases or decreases as the input increases, is a common phenomenon of many tasks in our daily lives. This can be easily observed in the straightforward example of pouring water from a teapot into a cup. In this scenario, as we pour water from the teapot, the tilt angle of the teapot decreases in a consistent and steady manner, while the volume of water in the cup increases correspondingly. Another example of monotonicity can be observed when we try to straighten a rope. To accomplish this task, we will hold the two ends of the rope and pull them apart. As we do this, the distance between the two ends will also increase in a steady and consistent manner. This type of consistent and controlled environment, in which a system's output always increases or decreases with a specific input, is referred to as a **monotonic physical system** (see Section 3.2).

Motivated by the above intuition of monotonicity, in this paper, we propose a novel framework, the **P**artial **O**rder **R**epresentation (POR), as a solution to the sample complexity problem in monotonic physical systems. Our proposed approach improves the learning process of monotonic quantities by incorporating an additional auxiliary loss term, which encourages the encoder to learn the implicit partial order in the environment. The POR framework enables the reinforcement learning algorithm to more effectively capture representations of the environment's monotonicity, such as the water level when pouring water, which may provide valuable signals for the RL algorithm during training. We showcase the effectiveness of the POR framework by evaluating it on the Softgym benchmark Lin et al. (2020), which comprises a variety of challenging visual-motor control tasks in monotonic

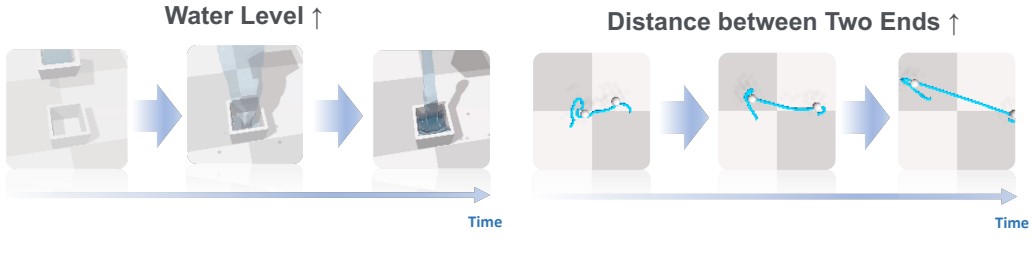

(a) Pour water into a target container.      (b) Straighten a rope.

Figure 1: A couple of examples of monotonic physical systems: **(a) Pour water into a target container**: When pouring water, the water level in the target container will not decrease. **(b) Straighten a rope**: When straightening a rope, the distance between two ends will not decrease.

physical systems. Our results demonstrate that the POR framework outperforms state-of-the-art (SOTA) methods regarding performance and sample efficiency on a diverse set of challenging visual-motor control tasks. To further understand the features learned by POR, we also present visualizations of the feature map.

**The main contribution of our work is as follows:**

- Our findings indicate that solving the sample complexity problem is essential for achieving success in complex high-dimensional visual-motor control tasks using model-free RL for tasks with monotonic physical systems. Furthermore, our research has uncovered that the monotonicity inherent in these systems offers valuable signals to the RL algorithm.
- We propose a novel POR framework, which utilizes the monotonicity property to improve the state representation learning of the environment.
- We verify the superiority of POR by a variety of challenging visual-motor control tasks, and the experimental results demonstrate that POR achieves superior performance and sample efficiency compared to state-of-the-art methods.

## 2 RELATED WORKS

### 2.1 LEARNING MONOTONIC PHYSICAL SYSTEMS

**Model-based approaches** To learn the monotonic physical systems, several works including Do & Burgard (2018); Rozo et al. (2013); Yamaguchi & Atkeson (2016); Yamakawa et al. (2012); Allali et al. (2017) apply model-based approaches. Some of these works, such as Do & Burgard (2018); Yamakawa et al. (2012); Yamaguchi & Atkeson (2016); Pan et al. (2016); Allali et al. (2017), approximate the model dynamics using direct algebraic calculations. On the other hand, Rozo et al. (2013); Gondry et al. (2019) consider using hidden Markov models to model the dynamics. Once the dynamics of the environment have been modeled, this approach directly applies optimization-based motion planning to their tasks.

Besides modeling with classical methods, recently a branch of work try to use neural networks to learn the dynamics. Several works, such as Li et al. (2022); Ma et al. (2021); Yan et al. (2021); Hafner et al. (2019), have explored model-based reinforcement learning. These approaches aim to learn models for dynamic 3D scenes purely from 2D visual observations using neural networks, graph neural networks or mesh-based neural networks, etc. These lines of work improve the generalizability of pure analytical methods. However, they still need to learn a relatively heavy model and pose a computational burden on real-time operation devices.

**Model-free approaches** Despite the advances in model-based approaches, recent studies by Zhu et al. (2021); Chen et al. (2020); Lin et al. (2020) have shown that directly applying model-free reinforcement learning to these applications can be challenging. It is difficult to learn a useful policy due to high dimensional complex system dynamics. To alleviate this issue, Salhotra et al. (2022);

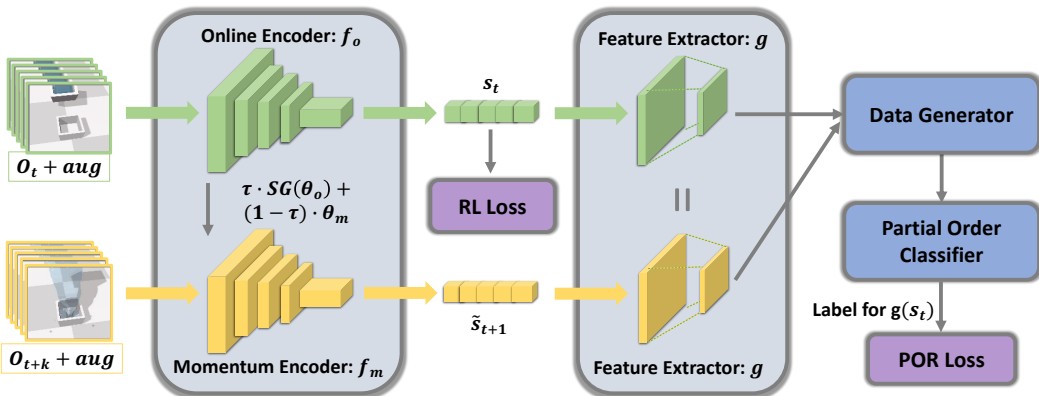

Figure 2: An illustration of the POR framework. The POR framework has four main components: **(a) Online and Momentum Encoder** provide a state representation from an observation. **(b) Feature Extractor** retrieves the monotonicity from the state representation. **(c) Data Generator** transforms the features into the input form of the next component. **(d) Partial Order Classifier** is used to distinguish the order of partial order.

Vecerik et al. (2017) consider using expert demonstrations as an additional signal to learn a good policy, which provides useful guidance. Another line of works including Lin et al. (2022b); Xu et al. (2022) uses specifically-designed architectures to solve problems like cloth manipulation. Niu et al. (2023) considers the goal distributions to create a curriculum throughout the learning process. These architectures, though achieve significant performance gains, are often highly specialized to the tasks at hand and can hardly generalize to other applications.

## 2.2 MODEL-FREE REINFORCEMENT LEARNING FOR VISUAL-MOTOR CONTROL

We evaluate the efficacy of various algorithms in terms of their sample efficiency and performance on the Softgym benchmark (Lin et al., 2020). The Softgym benchmark has gained widespread popularity among researchers in the field, including Laskin et al. (2020); Haarnoja et al. (2018); Kostrikov et al. (2020). These model-free RL studies have used Softgym to compare the sample efficiency of different image-based visual-motor control tasks. Furthermore, Schwarzer et al. (2020) have proposed learning self-predictive representations to improve the sample efficiency of algorithms for model-free RL.

## 3 METHOD

We propose the Partial Order Representation (POR) framework for reinforcement learning. POR could be implemented as an add-on component for existing RL algorithms by training a partial order classifier as an auxiliary loss. POR facilitates learning representations considering the environment's monotonicity, which provides valuable signals for RL algorithms during training. In this section, We will describe the proposed method in detail.

### 3.1 PROBLEM FORMULATION

This work investigates image-observation Reinforcement Learning (RL) problems within the Partially Observable Markov Decision Process (POMDP) formulation. A POMDP is represented by a 7-tuple $(\mathcal{S}, \mathcal{A}, T, R, \Omega, O, \gamma)$, where $\mathcal{S}$ is the state space, $\mathcal{A}$ is the action space, $T$ is the conditional transition probabilities between states, $R$ is the reward function, $\Omega$ is the observation space, $O$ is the conditional observation probabilities, and $\gamma$ is the discount factor.

At time step $t$, the environment is in state $s_t \in \mathcal{S}$. The agent takes an action $a_t \in \mathcal{A}$, which causes the environment to transit to state $s_{t+1}$ with probability $T(s_{t+1}|s_t, a_t)$. Subsequently, the agent receives a new observation $o_{t+1} \in \Omega$. The observation depends on the new state of the environment $s_{t+1}$ and is sampled from $O(o_{t+1}|s_{t+1})$. Finally, the agent receives a reward $r_t = R(s_t)$.

In a POMDP, an agent interacts with the environment through a sequence of actions in each episode. Our goal is to train the agent to maximize the expected cumulative rewards within an episode, that is $\mathbb{E}[\sum_{t=0}^{\infty} \gamma^t r_t]$.

## 3.2 Monotonic Physical System

In this section, we give the definition of monotonic physical systems. In this work, we focus on deterministic dynamics since most physical dynamics are intrinsically deterministic. The corresponding transition probability can be defined as follows:

$$T(s_{t+1}|s_t, a_t) = \begin{cases} \infty, & \text{if } s_{t+1} = h(s_t, a_t) \\ 0, & \text{otherwise} \end{cases} \tag{1}$$

where $h$ is the deterministic dynamics function, $s_t, s_{t+1} \in \mathcal{S} \subseteq \mathbb{R}^n$ are the states at time step $t$ and time step $t+1$ respectively, $a_t \in \mathcal{A} \subseteq \mathbb{R}^q$ is the action at time step $t$, $n$ is the dimension of the states, and $q$ is the dimension of the actions.

We also need to formally define partial order $\preceq$, which is used to compare two vectors, $x, x'$, of the same dimension.

$$x \preceq x' \Leftrightarrow x_i \leq x'_i, \forall i = 1, \ldots, n, \tag{2}$$

where $x_i, x'_i \in \mathbb{R}$ denote the $i$-th entry of the vectors $x$ and $x'$ respectively. This equation states that for two vectors $x$ and $x'$, $x$ is less or equal to $x'$ if and only if the $i$-th entry of $x$ is less or equal to the $i$-th entry of $x'$ for all $i$ between 1 and $n$.

Now we give the definition of monotonic physical systems.

**Monotonic Physical System.** A discrete-time dynamical system is a monotonic physical system if there exists a function $g : \mathbb{R}^n \to \mathbb{R}^m, n > m$, such that for any $s_t \in \mathcal{S}$, there exists an action $a_t \in \mathcal{A}$ leads to $g(s_t) \preceq g(s_{t+1})$.

Let us demonstrate a couple of intuitive examples of monotonic physical systems.

- **Pouring water from a teapot into a cup** In this scenario, the state $s$ can include the positions and orientations of water particles, a teapot, and a cup. An essential feature, represented by the mapping function $g$, is the upward movement of the average mass center of the water particles and the orientation of the teapot's spout. Through examination of these physical properties, it is clear that this system adheres to our definition of a monotonic physical system, as $g(s)$ increases.

- **Straightening a rope** The state $s$ in this task includes the positions and orientations of the rope particles, as well as the left-hand and right-hand key points. This system is a monotonic physical system because the distance between the particle at the end of the rope gradually increases. This can be represented by a mapping function $g$, that focuses on the physical properties of the system.

## 3.3 Partial Order Representations

The architecture of the Partial Order Representation (POR) framework is illustrated in Figure 2. The primary goal of POR is to improve the performance and the data efficiency of RL algorithms in monotonic physical systems by encouraging state representations to capture the physical quantities in the environment that change monotonically over time. To achieve this goal, we design a classifier to determine the order relationship of these quantities.

The POR framework has four main components, which are described as follows:

- **Online and Momentum Encoder:** Firstly, we use data augmentation to augment each observation independently. The current observation $o_t$ is used as the input of the online

encoder $f_o$, and the resulting latent vector is $s_t = f_o(o_t)$. At the same time, the next observation $o_{t+1}$ is processed by the momentum encoder, which is not updated through gradient descent, resulting in the latent vector $\tilde{s}_{t+1} = f_m(o_{t+1})$. To update the momentum encoder, we follow the prior work(He et al., 2020) and use an exponential moving average (EMA) method. Specifically, we denote the parameters of $f_o$ as $\theta_o$ and those of $f_m$ as $\theta_m$. The EMA method is applied to these parameters in order to update the momentum encoder. The use of data augmentation and the EMA method aims to improve the generalization of the encoder and reduce overfitting. The update rule for $\theta_m$ is:

$$\theta_m \leftarrow \tau\theta_o + (1-\tau)\theta_m \tag{3}$$

where $\tau \in [0, 1)$ is the EMA coefficient.

- **Feature Extractor:** This component, known as the mapping function $g : \mathbb{R}^n \to \mathbb{R}^m, m < n$, is utilized to extract specific physical quantities that exhibit monotonous changes in the environment. As a result of this step, we have $x_t = g(s_t) \preceq x_{t+1} = g(\tilde{s}_{t+1})$, where $x_t$ and $x_{t+1}$ represent the extracted physical quantities at time $t$ and $t + 1$, respectively.

- **Data Generator:** The Data Generator component is responsible for transforming the features extracted in the previous step into a format suitable for the classifier input. In particular, the Data Generator generates two instances, $(x_t, x_{t+k}, 0)$ and $(x_{t+k}, x_t, 1)$, for classification in the format (input1, input2, label), where input 1 and input 2 are the extracted physical quantities at two consecutive time steps. The label is either 0 or 1, indicating the partial order relationship of the input instances. This process is known as data preprocessing, it allows the classifier to train on a dataset that is well-structured, clean, and ready to be consumed by the model. In our design, we choose $k = 1$.

- **Partial Order Classifier:** The main function of this component is to perform binary classification by utilizing the comparability of partial order. Specifically, it is used to determine which of the two input elements is smaller. This can be done by comparing the physical quantities extracted by the mapping function $g$ and determining their relative order. The loss function used to train this component is designed to measure the difference between the predicted order and the true order of the input elements, and it corresponds to the binary cross entropy (BCE) function and is defined as follows:

$$L_\theta^{\text{POR}} = -\log(1 - c(x_t, x_{t+1})) - \log(c(x_{t+1}, x_t)) \tag{4}$$

where $c$ is the partial order classification function.

During the training process, the model combines two different loss functions in order to update the online encoder: the POR loss $L_\theta^{\text{POR}}$ and the RL loss $L_\theta^{\text{RL}}$. The POR loss affects the parameters of the mapping function $g$, the partial order classification function $c$, and the online encoder $f_o$. It is designed to measure the difference between the predicted order and the true order of the input elements and is calculated by using the BCE loss function. On the other hand, the RL loss affects the online encoder $f_o$. It is used to optimize the policy for selecting the next instance to be processed and is calculated by using the reinforced learning algorithm. The full optimization objective is a weighted sum of these two loss functions, $L = \lambda \cdot L_\theta^{\text{POR}} + L_\theta^{\text{RL}}$, where $\lambda$ is a scalar value that determines the relative importance of each loss function. For more implementation details, please see Appendix A.1 and Appendix A.2.

## 4 EXPERIMENTS

In our study, we employed a variety of challenging visual-motor control tasks to evaluate the performance of our proposed method. The results of our experiments indicate that POR outperforms the baselines in terms of performance and sample efficiency. Specifically, POR was able to achieve higher performance and sample efficiency. These findings demonstrate the effectiveness and efficiency of POR as a method for solving visual-motor control tasks.

### 4.1 ENVIRONMENTS

We chose Softgym (Lin et al., 2020) as our benchmark because it has more noticeable monotonicity compared to other environments. Additionally, Softgym offers complex task environments that

---

**Algorithm 1** **P**artial **O**rder **R**epresentation

---

Denote parameters of online encoder $f_o$ as $\theta_o$.
Denote parameters of momentum encoder $f_m$ as $\theta_m$.
Denote parameters of feature extractor $g$ and partial order classification model $c$ as $\phi$.
Denote parameters of RL model as $\omega$.
Denote the maximum interval as $K$ and the batch size as $N$.
initialize replay buffer $B$.
**while** $Training$ **do**
    collect experience $(o, a, r, o')$ with $(\theta_o, \omega)$ and add to buffer $B$.
    sample a minibatch of sequence of $(o, a, r, o') \sim B$.
    $D \leftarrow empty$
    **for** $i$ **in** $range(0, N)$ **do**
        **if** $augmentation$ **then**
            $o^i \leftarrow \text{augment}(o^i); o'^i \leftarrow \text{augment}(o'^i)$
        **end if**
        $s^i \leftarrow f_o(o^i); \tilde{s}'^i \leftarrow f_m(o'^i)$
        $x^i \leftarrow g(s^i); x'^i \leftarrow g(\tilde{s}'^i)$
        $l^i \leftarrow -\log(1 - c(x^i, x'^i)) - \log(c(x'^i, x^i))$
        $l^i \leftarrow \lambda l^i + \text{RL loss}(o_t^i, a, r, o_{t+1}^i; \theta_o)$
    **end for**
    $l \leftarrow \frac{1}{N} \sum_{i=0}^N l^i$
    $\theta_o, \phi, \omega \leftarrow \text{optimize}((\theta_o, \phi, \omega), l)$
    $\theta_m \leftarrow \tau\theta_o + (1 - \tau)\theta_m$
**end while**

---

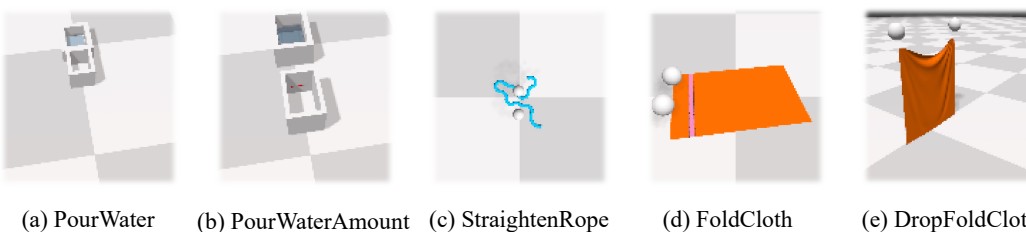

(a) PourWater    (b) PourWaterAmount    (c) StraightenRope    (d) FoldCloth    (e) DropFoldCloth

Figure 3: Selected tasks in Softgym: (a) PourWater, (b) PourWaterAmount, (c) StraightenRope, (d) FoldCloth, and (e) DropFoldCloth.

challenge state representation, revealing performance differences among algorithms. It contains benchmarks designed to simulate complex, deformable manipulation tasks. It encompasses a wide range of challenging visual robotic tasks with continuous action spaces. As depicted in Figure 3, Softgym includes a diverse set of tasks that involve interacting with deformable objects such as ropes, cloth, and fluid. We present the normalized performance scores for each task, as provided by the benchmark's environments. These scores offer a standard way to evaluate and compare the performance of different methods on various tasks.

We compare the effectiveness of POR to several baselines in the field. These include:

(1) **SPR**(Schwarzer et al., 2020), a previous state-of-the-art approach for achieving temporal consistency through the use of data augmentations and a multi-step consistency loss using the BYOL-style(Grill et al., 2020) self-supervised learning.

(2) **DrQ**(Kostrikov et al., 2020), a method that enhances the input images through data augmentations while simultaneously learning the original reinforcement learning objective.

(3) **CURL**(Laskin et al., 2020), a technique that utilizes contrastive learning as a supplementary task to improve the quality of the image representations.

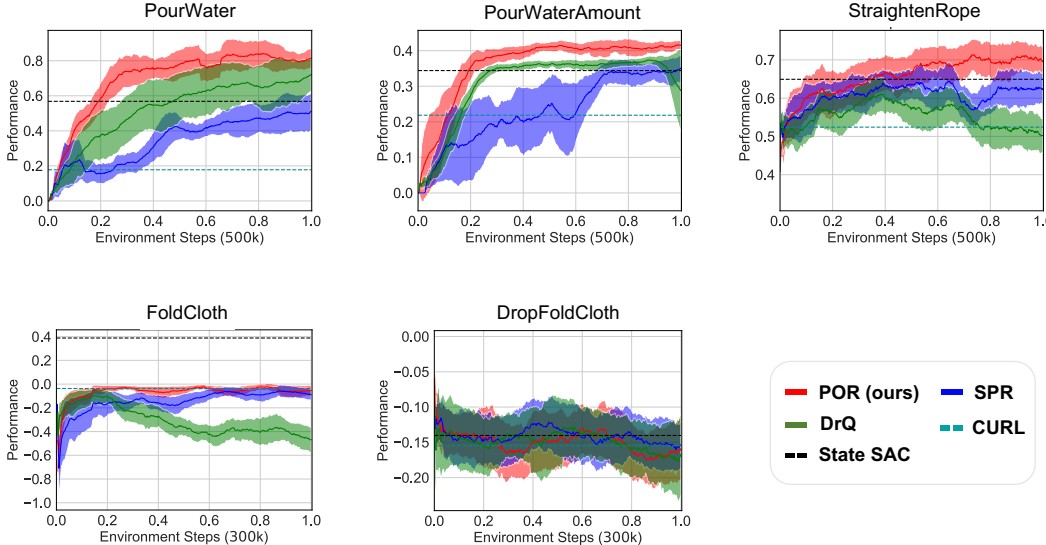

Figure 4: Performance on Softgym benchmark. The shaded areas show the standard deviation across 3 seeds. POR achieves better performance and sample efficiency in various challenging tasks.

(4) **State SAC**(Haarnoja et al., 2018), an approach that applies the Soft Actor-Critic algorithm directly to low-dimensional states derived from the ground truth, rather than the raw pixel data.

The following will give a brief description of the tasks we have chosen, as shown in Figure 3.

- **PourWater**: Pour a cup of water into a target cup. The reward is proportional to the water level of the target.
- **PourWaterAmount**: This task is similar to PourWater but requires a specific amount of water poured into the target cup. The required water level is indicated by a red line.
- **StraightenRope**: Straighten a rope starting from a random configuration. The reward is proportional to the distance between the ends of the rope.
- **FoldCloth**: Fold a piece of flattened cloth in half. Particles are divided into two groups, $a$ and $b$. The reward will be the negative average Euclidean distance between each particle in group $a$ and the corresponding particle in group $b$.
- **DropFoldCloth**: This task is similar to FoldCloth, but the initial state is in the air.

## 4.2 RESULTS

In our experimental studies, we assess the performance of POR using a variety of tasks within the Softgym environment. These tasks have been specifically designed to encompass all of the different types of deformable objects presented in Softgym. Results are shown in Figure 4.

POR exhibits impressive sample efficiency on the PourWater and PourWaterAmount tasks, achieving a 1.32x and 1.11x improvement in efficiency at 200k steps when compared to other image input baselines. For StraightenRope and FoldCloth tasks, POR has 1.04x and 1.02x improvement.

Additionally, at 500k steps, POR achieves superior performance compared to all baselines on the PourWater, PourWaterAmount, and StraightenRope tasks.

It is only outperformed by State SAC on the FoldCloth task while achieving comparable performance to all the baselines on the DropFoldCloth task at 300k steps. Although POR outperforms compared to other visual input methods, it still does not match the performance of the state input method in the FoldCloth task, a state unattainable in real-world settings.

It is clear to find that if there is obvious monotonicity, POR can effectively improve sample efficiency. On the contrary, for tasks that do not have obvious monotonicity, it will not have a negative impact on sample efficiency.

### 4.3 CASE STUDY

From the results, we can see that POR has significantly better performance and sample efficiency than baselines. However, to determine the effectiveness of POR in aiding the learning of monotonic physical quantities within an environment, we will analyze the auxiliary loss generated by POR. To do this, we will divide the environment into two distinct categories by the loss curve: (a) Monotonic Objective, (b) Non-monotonic Objective.

**Monotonic Objective** The monotonic objective here refers to the existence of monotonicity in the environment is positively correlated with the objective (reward) of the task. As depicted in Figure 5, the convergence of POR loss is demonstrated over a period of 300k steps. It is clearly evident that the POR loss successfully converges in this type of environment, with a small standard deviation. This indicates that the classifier can precisely identify the implicit partial order relationship within the environment, and this is independent of the action sequence.

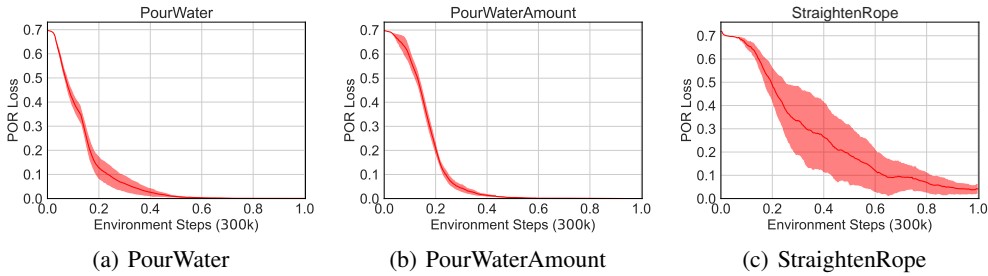

(a) PourWater    (b) PourWaterAmount    (c) StraightenRope

Figure 5: POR Loss Curve for Monotonic Objective at 300k steps with 3 different seeds.

**Non-monotonic Objective** Here, non-monotonic objective refers to the following two specific types:

- There is no direct correlation between the presence of monotonicity in the environment and the objective (reward) of the task. As illustrated in figure Figure 6(a), at 300k steps, the POR loss is approximately 0.5, and the accuracy rate for the classification task being performed is around 60%. Therefore, in this type of environment, the POR algorithm is unable to effectively learn and extract the hidden information that pertains to the partial order relationship present in the environment.
- Environments with salient monotonicity. For example, when a ball is dropped from a tall building, its vertical position (ordinate) will decrease over time as it falls due to gravity. As depicted in figure Figure 6(b), due to the influence of gravity in this environment, the POR loss converges at a faster rate and with a smaller standard deviation.

### 4.4 VISUALIZATION

To better understand what the partial order classifier has learned, we can generate a saliency map of the features and visualize them, as shown in Figure 7.

The saliency map highlights the parts of the features that have a higher weight in the neural network. The brighter areas in the map correspond to the features that the model believes it is more important in making the predictions. It is apparent that the POR for the PourWater task is primarily determined by the quantity of water present in the target cup. In the case of the StraightenRope task, the POR's evaluation of the partial order is influenced by the position of the particles throughout the rope. However, it is evident that the brightness of the saliency map is higher at both ends of the rope compared to other positions.

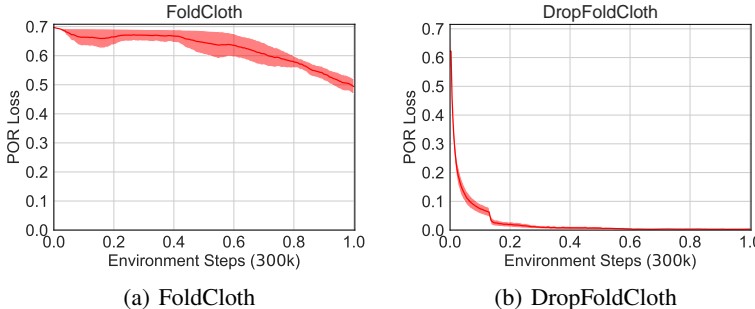

(a) FoldCloth          (b) DropFoldCloth

Figure 6: POR Loss Curve for Non-monotonic Objective at 300k steps with 3 different seeds.

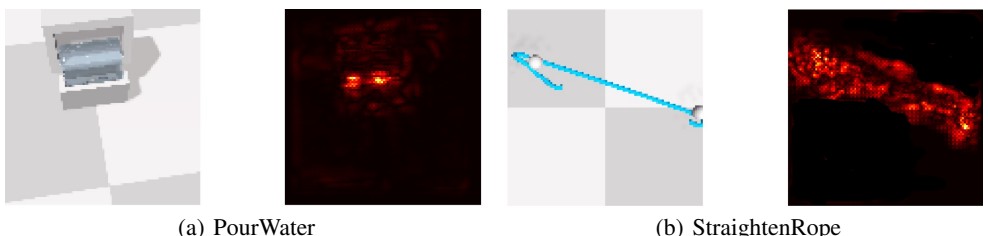

(a) PourWater          (b) StraightenRope

Figure 7: Saliency Map for PourWater and StraightenRope tasks. The saliency map provides a visualization of the features that the POR has learned to be significant in predicting partial orders for the PourWater and StraightenRope tasks. The brighter areas in the map indicate the parts of the features that are deemed more critical by the model in making its predictions.

## 5   LIMITATION

Although POR has better or comparable sample efficiency than all visual input baselines in all tasks. However, for some tasks, such as FoldCloth, like other visual input baselines, it is difficult to obtain a better encoder. This results in poorer performance than state SAC. In addition, this work is now primarily based on empirical results, we currently do not have a theoretical explanation to prove the effectiveness of our POR framework.

## 6   CONCLUSION

Prior model-free reinforcement learning techniques may face challenges when dealing with complex high-dimensional visual-motor control tasks. This paper proposes a new framework, called **P**artial **O**rder **R**epresentation (POR), to address the sample complexity issue in monotonic physical systems. We demonstrate that many systems have implicit physical properties, such as monotonicity, that can benefit learning algorithms. However, these properties have been overlooked by previous work. The POR framework allows the reinforcement learning algorithm to capture the environment's monotonicity and use it as a signal during training. We evaluate the effectiveness of POR on the Softgym benchmark, which includes various challenging visual-motor control tasks in monotonic physical systems. Our results, presented in Section 4.2, indicate that POR outperforms state-of-the-art methods in terms of performance and sample efficiency across a range of challenging visual-motor control tasks. We also classify the experimental tasks and provide an analysis of our method for each classification, as presented in Section 4.3 and 4.4. However, while this work is primarily based on empirical results, we plan to explore a more theoretical explanation for the effectiveness of POR in future work.

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
