# A APPENDIX

## A.1 NETWORK ARCHITECTURE

We have developed a network architecture that uses the soft actor-critic algorithm as the reinforcement learning backbone. Our architecture is based on the implementation presented in Lin et al. (2020), with the addition of a POR component, as illustrated in Figure 2. For the feature extractor, we employ a multi-layer perceptron (MLP) with three hidden layers of sizes $256, 256$, and an output layer of size $32$. We set the momentum to $0.1$ to improve convergence. For the partial order classifier, we also use an MLP, with five hidden layers of sizes $128, 128, 64, 32, 16$, and an output layer of size $1$. The momentum is set to $0.1$. The classifier is responsible for predicting the correct POR label for each sample. By adding the POR component, our architecture can better capture the monotonic properties of the physical system and improve the sample efficiency and performance.

## A.2 HYPERPARAMETERS

The general hyper-parameters for POR can be found in Table 1.

| Parameter | SAC |
| --- | --- |
| batch size | 128 |
| initial steps | 1000 |
| replay buffer size | 1e5 |
| target smoothing coefficient | 0.01 |
| alpha | automatic tuning |
| delayed policy update period | 2 |
| target update interval | 2 |

Table 1: General hyper-parameters for POR.

To achieve learning stability, we tuned the reward scaling and learning rate for POR, for each environment:

- PourWater: learning rate = 3e-4, reward scaling = 20, no learning rate decay.
- PourWaterAmount: learning rate = 3e-4, reward scaling = 20, no learning rate decay.
- StraightenRope: learning rate = 3e-4, reward scaling = 50, have learning rate decay.
- FoldCloth: learning rate = 1e-4, reward scaling = 50, no learning rate decay.
- DropFoldCloth: learning rate = 3e-4, reward scaling = 50, no learning rate decay.