# OpenReview forum: "Reinforcement Learning with Partial Order Representation for Monotonic Physical System"
_ICLR.cc/2024/Conference — ICLR 2024 Conference Withdrawn Submission_

### Official Review · Reviewer_T6bZ · 2023-10-27

**Soundness:** 1 poor
**Presentation:** 3 good
**Contribution:** 1 poor
**Rating:** 3
**Confidence:** 5

**Summary:**

In this paper, the authors propose to use a notion of system monotonicity to improve  reinforcement learning performance, in terms of  enhanced performance and reduced sample complexity.

The authors propose aPartial Order Representation (POR) framework, with the aim to improve the ability of reinforcement learning to capture the systems’ monotonicity, resulting in valuable signals during training that can enhance performance and reduce sample complexity.

**Strengths:**

The paper has an interesting task, and proposes a particular environment, SoftGym, to study this task. There is clear novelty. The stated goals (at a high level) are clear.

**Weaknesses:**

I am concerned by the lack of technical depth to the notion of monotone systems, and in the claims of systems being monotone. It is simple to define counter-examples to the definition of monotonicity of systems, and hence everything else in the article falls apart. It is difficult to see how the experiments can be definitive if the basic definitions and the claims onf monotone behaviours of the systems for experiments are not clearly following the desired properties.



There is significant work on monotone systems---please provide the normal references, e.g.,

Hirsch, M. W., & Smith, H. (2006). Monotone dynamical systems. Handbook of differential equations: ordinary differential equations, 2, 239-357.

D. Angeli and E. D. Sontag. Monotone control systems. IEEE Trans. Autom. Control, 48(10):1684–1698, October 2003.

The definition of monotone system contradicts the standard definition.
"Monotonic Physical System. A discrete-time dynamical system is a monotonic physical system if there exists a function g : Rn → Rm, n > m, such that for any s_t ∈ S, there exists an action at ∈ A
leads to g(st) >= g(st+1)."

In the definition it notes "for any s_t ∈ S": do you mean "for *every* state"?
By the given definition there may be only *one* state that is montone, and all others are non-monotone. This does not make sense.

The definition of monotone control system is much more precise:

for control space U, state space X, and parameter Ω, it must hold that:

u1 ⪰ u2,ω1 ⪰ ω2 ⇒ ϕ(t, x1,ω1,u1) ⪰ ϕ(t, x2,ω2,u2), for u1,u2 ∈ U, ω1,ω2 ∈ Ω, x1, x2 ∈ X.

There is no proof or precise technical demonstration that the systems studied are in fact monotone: it is only a hand-waving argument.

"Through examination of these physical properties, it is clear that this system adheres to our definition of a monotonic physical system, as g(s) increases."

---this is only a hand-waving argument---it is NOT clear to this reviewer. it is not even precise what is monotonic!

In sec. 4.3, I cannot understand the material following "Non-monotonic Objective: Here, non-monotonic objective refers"---very confusing!

The supplementary material is insufficient to provide a means to check the validity of the experiments. I need to see the underlying code to check if what is implemented is correct.

The results on sample efficiency are insufficient to justify the claims. For example, it is stated: "For StraightenRope and FoldCloth tasks, POR has 1.04x and 1.02x improvement." This is really not sigificant. There is too little data on multiple experiments to make the strong claims noted.

**Questions:**

1. Can you compare your definition of monotonicity to that of the literature, and show that yours is correct?
2. Can you provide precise specifications of the experiemental domains to show that they adhere to a correct definitoin of monotonicity?

---

### Official Review · Reviewer_Beac · 2023-10-30

**Soundness:** 4 excellent
**Presentation:** 3 good
**Contribution:** 4 excellent
**Rating:** 5
**Confidence:** 5

**Summary:**

This paper proposes `Partial Order Representation (POR)` framework, building around capturing monotonicity. The core idea is that many physics-based systems inherently display specific physical characteristics that are advantageous for learning algorithms. One such characteristic is the monotonicity of the environment. Monotonicity is defined as the consistent and gradual increase or decrease of a function or system's state variable based on the input. The task example of this characteristic considered here is when pouring water into a cup from a teapot, as the water is poured, the teapot's tilt angle consistently reduces, while the water volume in the cup correspondingly rises. Similarly, when straightening a rope by pulling its two ends, the distance between the ends consistently enlarges.

### Methodology
The aim is to teach a system to understand the natural order of events or states in the environment, in a sense to know if one state comes before or after another.

**Encoders**:
   - **Online Encoder**: Processes the current observation/state.
   - **Momentum Encoder**: Processes the next observation/state. It is called 'momentum' because instead of updating through regular backpropagation, it uses an exponential moving average (EMA) based on prior updates.

**Feature Extractors**:
   - After passing the observations through the encoders, we need to extract relevant features from the encoded states to further represent the monotonicity.
   - The feature extractor, represented by the mapping function, does this by converting the latent representation (from the encoders) into some specific physical quantities that demonstrate monotonic behavior.

**Data Generator and Classifier**:
   - Once the features are extracted, they are processed by the Data Generator, which prepares them for the classifier. It creates pairs of data points from two consecutive time steps.
   - The pairs are labeled either 0 or 1. The idea is to determine the partial order relationship between two states.
      - A label of `0` might mean that the first state $\( x_t \)$ comes before the second state $\( x_{t+k} \)$.
      - A label of `1` could imply the opposite, i.e., the state $\( x_{t+k} \)$ comes before $\( x_t \)$.

**Optimization**
- The model combines two loss functions the `POR` and `RL` loss.

### Experiments
Experiments include : `Softgym benchmark Pouring water from a teapot into a cup` & `Straightening a rope`. Baselines are `DrQ`, `SPR` and `CURL`.

**Strengths:**

- The idea of learning the monotonicity property based on partial order of states is interesting and novel.
- By using the classifier to predict these labels, the system learns the order or sequence in which states/observations follow each other in the environment. This is crucial for understanding monotonic systems, where some quantities consistently increase or decrease.
- In essence, the system is being trained to recognize if one observation (or the physical quantity derived from it) naturally comes before or after another. This could be analogous to teaching a system to recognize that dawn comes before morning or that a glass being filled with water will have a higher water level as time progresses.
- Ablation study of the monotonic objective and the correlation with the reward task seems convincing to me.

**Weaknesses:**

- All the baselines comparisons in this paper are from 2020 - There has been a ton of development in representation learning for RL, I’m proposing a set of options just based on the recent SOT citations you can choose from to compare where appropriate (some are model-based, some are model-free). Please see references below.

- Overall I like the core idea presented in this paper, but I think comparing against more recent baselines, can strengthen this paper. The paper as it stands is not ready for publication so I highly recommend the authors to consider comparing against SOTs. Happy to re-consider my score if this can be incorporated during rebuttal.

[1] Mastering Visual Continuous Control: Improved Data-Augmented Reinforcement Learning - https://arxiv.org/abs/2107.09645

[2] Masked World Models for Visual Control - https://proceedings.mlr.press/v205/seo23a/seo23a.pdf

[3] Bigger, Better, Faster: Human-level Atari with human-level efficiency - https://proceedings.mlr.press/v202/schwarzer23a/schwarzer23a.pdf

[4] Mastering Diverse Domains through World Models - https://arxiv.org/pdf/2301.04104.pdf

[5] Stabilizing Deep Q-Learning with ConvNets and Vision Transformers under Data Augmentation - https://arxiv.org/pdf/2107.00644.pdf

[6] Improving Generalization in Visual Reinforcement Learning via Conflict-aware Gradient Agreement Augmentation - https://openaccess.thecvf.com/content/ICCV2023/papers/Liu_Improving_Generalization_in_Visual_Reinforcement_Learning_via_Conflict-aware_Gradient_Agreement_ICCV_2023_paper.pdf

[7] Visual Reinforcement Learning with Self-Supervised 3D Representations - https://arxiv.org/pdf/2210.07241.pdf

[8] REBOOT: Reuse Data for Bootstrapping Efficient Real-World Dexterous Manipulation - https://arxiv.org/pdf/2309.03322.pdf

**Questions:**

- Figure 4 `DropFoldCloth` seems to be failing completely  ? I'm not sure what information I can gather from this clustered performance of all baselines.

---

### Official Review · Reviewer_nyme · 2023-11-05

**Soundness:** 2 fair
**Presentation:** 3 good
**Contribution:** 2 fair
**Rating:** 5
**Confidence:** 4

**Summary:**

This paper introduces a new algorithm that aims to exploit any partial ordering properties that might be present in a task/environment (e.g. objects consistently fall due to gravity). The proposed algorithm aims to exploit such regularities in the dynamics and does so by introducing a new loss function to model-free RL via the use of a partial ordering classifier in a learned feature space. The resulting method shows consistent albeit modest improvements on several RL environments.

**Strengths:**

- The paper motivates the core idea of the paper well. That said, I have some concerns about how broadly applicable the ideas are as mentioned in the weaknesses section.
- The method shoes consistent, albeit modest, performance gains across several environments.

**Weaknesses:**

- It is unclear to me if the monotonic property is a property of a physical system (e.g. an object) or that of a specific task. For example, the authors provide the example of "straightening a rope" in Figure 1. If the goal is to tie a knot on the same rope, it is unclear if any monotonic property is preserved. Based on this, I find the concept of a "monotonic physical system" to be somewhat misleading, since it also depends on the specifics of the task.
- In Eq (1), it seems odd to designate infinite probability to the deterministic transition. Perhaps it is better to define it as a delta distribution.
- In Section 3.3, there is significant overloading of notation between "notional" Markovian state in a POMDP and the learned feature representation. For example, $s_t = f_o(o_t)$ would not be possible in a true POMDP, since to recover the state the entire history may be necessary. Furthermore, the state representation in which a partial ordering is preserved (e.g. the ends of a rope in the rope straightening example) may not be the representation a network actually learns. It is important to consistently use $z_t = f_o(o_t)$ to avoid confounding between $z_t$ (learned features) and $s_t$ (the actual Markovian state).
- Why no comparisons to model-based algorithms (e.g. Dreamer, TD-MPC)? All the experimental comparisons are with model-free methods. It can be argued that POR is a pseudo-model-based method since it is trained to have a partial ordering between $x_t$ and $x_{t+1}$. It is well known that model-based methods are typically more sample efficient than model-free methods. Could it be that POR learns faster simply due to some pseudo-model-based elements in the algorithm? A concrete comparison to model-based methods would alleviate this concern.

**Questions:**

See weaknesses section. Please address all of them.

---

### Official Review · Reviewer_TCFd · 2023-11-07

**Soundness:** 2 fair
**Presentation:** 3 good
**Contribution:** 2 fair
**Rating:** 3
**Confidence:** 3

**Summary:**

The paper presents a novel framework for improving the efficiency and performance of reinforcement learning (RL) algorithms by leveraging the deterministic and monotonic nature of many physical systems. The core concept is to exploit the inherent order in such systems, enabling more structured and informative state representations for RL tasks.

Central to the paper is the definition of monotonic physical systems, where state transitions are predictable and exhibit a clear, ordered relationship as a result of actions taken within the environment. The authors propose the Partial Order Representation (POR) framework which consists of:

An Online and Momentum Encoder that processes observations with the goal of reducing overfitting through an Exponential Moving Average (EMA) method.
A Feature Extractor that identifies and extracts physical quantities with monotonic properties from the environment's states.
A Data Generator that constructs training instances reflecting the monotonic relationships identified by the feature extractor.
A Partial Order Classifier employs binary classification to determine the relative order of the extracted features, providing a basis for the RL algorithm to understand the consequences of actions in a structured manner.
The paper integrates these components within an RL setting, employing two distinct loss functions: one that guides the partial order classification (POR loss) and another that optimizes the RL policy (RL loss). The combination of these losses is tuned by a weighting factor, which balances the contribution of each to the model's learning process.

The framework's efficacy is demonstrated through experiments on the Softgym benchmark, chosen for its complex, yet monotonically physical, tasks. The results suggest that the POR framework significantly outperforms traditional RL methods in both performance and sample efficiency, highlighting the benefits of incorporating structured, physical knowledge into learning algorithms.

**Strengths:**

The paper is well-motivated, identifying the need to exploit the structure within physical systems to improve RL outcomes.

It introduces a novel approach that integrates feature extraction, data generation, and partial order classification into the RL domain.

The empirical results suggest that the POR framework can lead to significant improvements in RL efficiency and performance.

**Weaknesses:**

The paper focuses on a very niche problem. I would advise the authors to give more examples of monotonic systems and show results on that. Perhaps a benchmark of similar tasks and not just the tasks from soft-gym environment.

Perhaps a better way to tackle these tasks is imitation learning with techniques like Diffusion Policy (https://arxiv.org/abs/2303.04137).

No real world results is another issue.

The dependence on the accurate definition and extraction of monotonic features may limit the framework's generalizability.

**Questions:**

It would be beneficial to understand the sensitivity of the method to the choice of the weighting factor λ in the combined loss function.